# Methyl radical chemistry in non-oxidative methane activation over metal single sites

Xin Huang[1,6], Daniel Eggart[2,6], Gangqiang Qin[1,3,6], Bidyut Bikash Sarma [4], Abhijeet Gaur [2], Jiuzhong Yang [5], Yang Pan [5], Mingrun Li[1], Jianqi Hao[1], Hongfei Yu[1], Anna Zimina[4], Xiaoguang Guo[1], Jianping Xiao [1], Jan-Dierk Grunwaldt [2,4] ✉, Xiulian Pan [1] ✉ & Xinhe Bao [1] ✉

Molybdenum supported on zeolites has been extensively studied as a catalyst for methane dehydroaromatization. Despite significant progress, the actual intermediates and particularly the first C-C bond formation have not yet been elucidated. Herein we report evolution of methyl radicals during non-oxidative methane activation over molybdenum single sites, which leads selectively to value-added chemicals. *Operando* X-ray absorption spectroscopy and online synchrotron vacuum ultraviolet photoionization mass spectroscopy in combination with electron microscopy and density functional theory calculations reveal the essential role of molybdenum single sites in the generation of methyl radicals and that the formation rate of methyl radicals is linearly correlated with the number of molybdenum single sites. Methyl radicals transform to ethane in the gas phase, which readily dehydrogenates to ethylene in the absence of zeolites. This is essentially similar to the reaction pathway over the previously reported $SiO_2$ lattice-confined single site iron catalyst. However, the availability of a zeolite, either in a physical mixture or as a support, directs the subsequent reaction pathway towards aromatization within the zeolite confined pores, resulting in benzene as the dominant hydrocarbon product. The findings reveal that methyl radical chemistry could be a general feature for metal single site catalysis regardless of the support (either zeolites MCM-22 and ZSM-5 or $SiO_2$) whereas the reaction over aggregated molybdenum carbide nanoparticles likely facilitates carbon deposition through surface C-C coupling. These findings allow furthering the fundamental insights into non-oxidative methane conversion to value-added chemicals.

Discovery of large reserves of shale gas, coal bed methane, and methane hydrate has reignited worldwide research interest in developing technologies to convert methane to transportable high-density energy sources or value-added chemicals, such as olefins and aromatics[1]. This can be achieved by well-developed indirect conversion routes via synthesis gas as platform followed by either Fischer-Tropsch synthesis, the newly developed OXZEO® or methanol chemistry[2–4]. The direct conversion route, which is in principle more

[1]State Key Laboratory of Catalysis, 2011-Collaborative Innovation Center of Chemistry for Energy Materials, Dalian Institute of Chemical Physics, Chinese Academy of Sciences, Dalian 116023, China. [2]Institute for Chemical Technology and Polymer Chemistry, Karlsruhe Institute of Technology, Karlsruhe 76131, Germany. [3]University of Chinese Academy of Sciences, Beijing 100049, China. [4]Institute of Catalysis Research and Technology, Karlsruhe Institute of Technology, Eggenstein-Leopoldshafen 76344, Germany. [5]National Synchrotron Radiation Laboratory, University of Science and Technology, Hefei 230029, China. [6]These authors contributed equally: Xin Huang, Daniel Eggart, Gangqiang Qin. ✉e-mail: grunwaldt@kit.edu; panxl@dicp.ac.cn; xhbao@dicp.ac.cn

energy- and cost-effective, is still under development[5]. Oxidative coupling of methane (OCM), methane dehydroaromatization (MDA), and the newly developed methane conversion to olefins, aromatics, and hydrogen (MTOAH) process over $SiO_2$-lattice confined single site iron catalysts Fe©$SiO_2$ have been demonstrated to show great potential[6]. Compared with oxidative activation, non-oxidative processes such as MDA and MTOAH could avoid over-oxidation and thus $CO_2$ emission in the reaction and consequently increase the carbon utilization efficiency[7–9].

Since Mo/ZSM-5 was first reported in 1993 for MDA reaction, a variety of other catalysts have been screened[10–12]. Among them, zeolite-supported Mo catalysts, such as Mo/ZSM-5 and Mo/MCM-22, were demonstrated to be still the most active so far[13]. Extensive efforts have been made in fundamental understanding of the nature of active sites and the reaction mechanism. There is a general consensus that $MoC_xO_y$ species forming on the Brønsted acid sites are the active sites for MDA reaction[14,15]. A bifunctional mechanism was generally accepted[16,17]: $CH_4$ is first activated on $MoC_xO_y$ sites, then the $C_2H_x$ species undergo oligomerization and aromatization over the adjacent Brønsted acid sites in zeolite micropores, forming mainly benzene, as well as a small amount of other aromatic products and even coke[18–20]. For instance, Lezcano-González et al. reported formation of $C_2H_x$/$C_3H_x$ over the metastable $MoC_xO_y$ species using in-situ high energy resolution fluorescence detected X-ray adsorption near-edge structure and X-ray emission spectroscopy after quenching the catalyst below 600 °C, during the MDA reaction at 677 °C[21]. On the other hand, Kosinov et al. recently proposed a hydrocarbon pool-like pathway, where activation of C−H bonds on Mo sites was followed by the transformation of (hydro)carbon intermediates confined inside the zeolite pores[22]. However, the initial C1 intermediates and the C-C coupling chemistry to $C_2H_x$ species are rarely studied and remain to be elucidated in both the bifunctional and hydrocarbon pool mechanisms. In this work, we dispersed Mo species on MCM-22 and found that non-oxidative methane activation is facilitated through methyl radical chemistry over Mo single sites followed by gas-phase C-C coupling forming ethane/ethylene, similar to the MTOAH reaction over $SiO_2$ lattice-confined iron single sites of Fe©$SiO_2$[23]. The reaction is moved towards aromatization only in the presence of zeolites, which gives benzene as the main product. Furthermore, our results indicate that the gas-phase methyl radical chemistry could be a general reaction feature for metal single-site catalysts.

## Results

### Methane conversion activity of xMo/MCM

To avoid aggregation of Mo species, Mo was dispersed with a loading lower than 1.0 wt.% on H-MCM-22 (Si/Al ratio = 16.5) using an incipient wetness impregnation method. The catalysts were named as xMo/MCM with x representing Mo loading (x = 0.25, 0.5, and 1.0; see Supplementary Table 1). Figure 1a and Supplementary Fig. 1 show that these catalysts exhibit typical MDA characteristics where the aromatic compounds dominate the hydrocarbon products. Within the induction period, the benzene yield increases with time on stream (TOS) to a maximum at about 70 min, and then decays gradually, consistent with previous studies (Supplementary Fig. 2)[24,25]. Furthermore, the 1Mo/MCM catalyst gives an initial $CH_4$ conversion of 17% at TOS = 10 min, reaching the thermodynamic equilibrium value at 750 °C (Supplementary Fig. 1). Interestingly, both $CH_4$ conversion and benzene formation rates (TOS = 70 min) show a linear trend with increasing Mo loading up to 1.0 wt.%, as shown in Fig. 1b. This hints at the essential role of the Mo sites for methane activation and benzene formation.

### Formation of methyl radicals

To capture reaction intermediates and to understand the reaction pathways, we employed synchrotron vacuum ultraviolet photoionization mass spectroscopy (SVUV-PIMS). This powerful technique allows detection of reactive, short-lived, and even gas-phase radical species, owing to the tunable photon energy and high energy resolution[26–28]. Figure 2a and Supplementary Fig. 3 show a signal of $m/z = 15$, in addition to 28, 40, 42, and 78 corresponding to stable products of ethylene, propyne, propylene, and benzene, respectively. The $m/z = 15$ signal is unambiguously attributed to the methyl radical according to its ionization threshold of 9.8 eV (Supplementary Fig. 4)[29–31]. This is similar to our previous observation of methane activation over the $SiO_2$ lattice confined iron single sites, Fe©$SiO_2$, where methyl radicals were also detected to be the reaction intermediates[23]. Interestingly, the relative concentration of both methyl radicals and ethylene correlate almost linearly with the Mo loading (Fig. 2b). Moreover, Fig. 2c shows that methyl radicals are continuously produced with time on stream. A temperature-programmed reaction depicted in Supplementary Fig. 5 proves that methyl radicals, $C_2H_4$, and $C_2H_6$ start to be detected at about 630 °C and the formation rate increases with rising temperature (Supplementary Figs. 3 and 6).

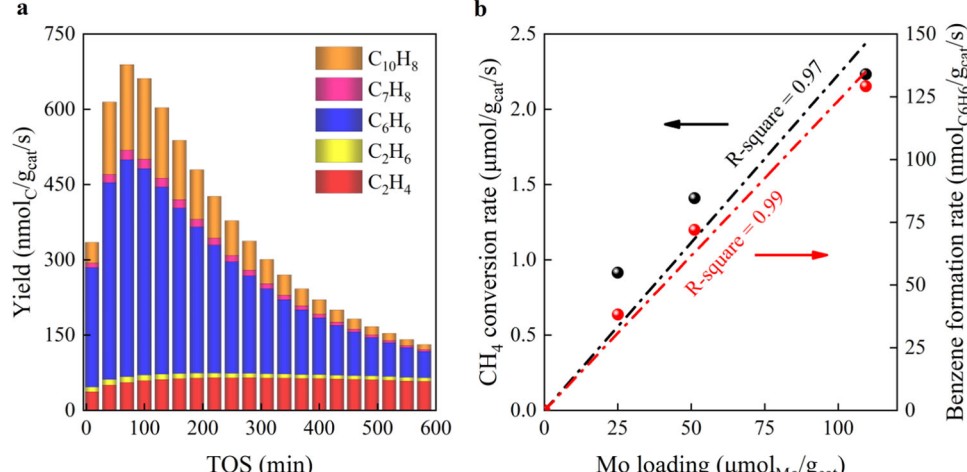

**Fig. 1 | MDA reaction over xMo/MCM catalysts. a** Yields of different products over 0.5Mo/MCM as a function of time on stream (TOS). **b** Benzene formation rate and corresponding $CH_4$ conversion rate as a function of Mo loading (TOS = 70 min). Reaction conditions: 750 °C, 0.1 MPa, 1.5 L/($g_{cat}$·h), and $CH_4$:$N_2$ = 9:1.

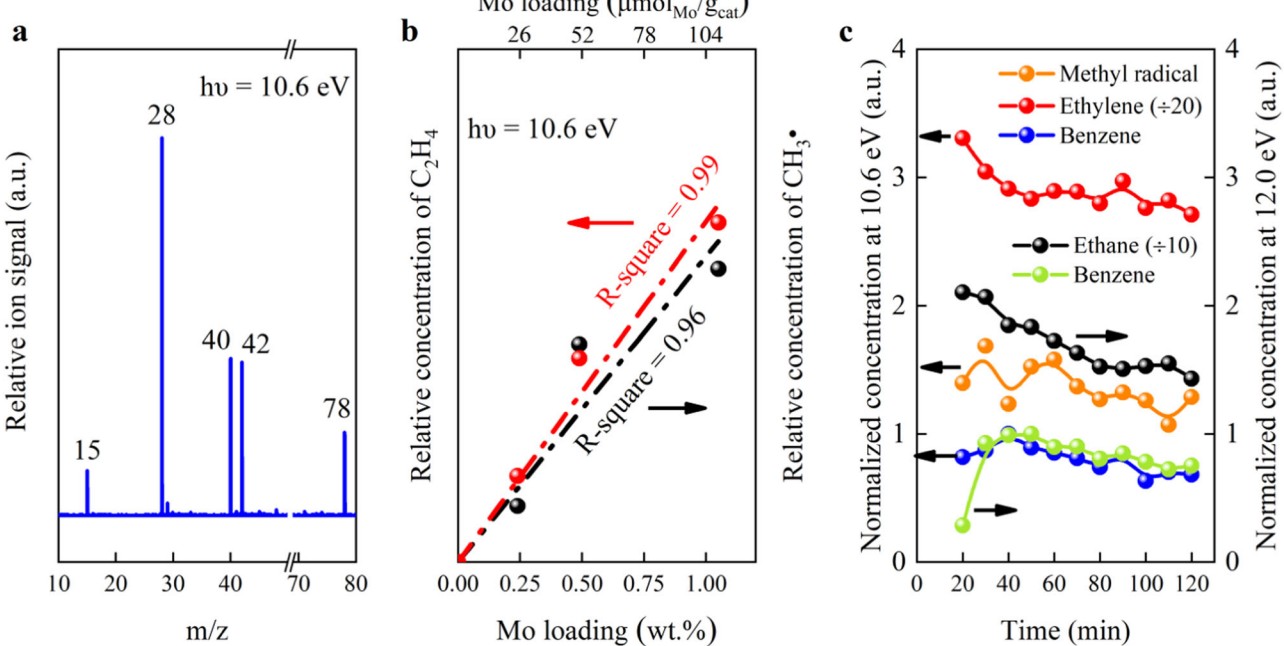

**Fig. 2 | Online SVUV-PIMS detection of the effluents over 0.5Mo/MCM. a** SVUV-PIMS spectrum. **b** Relative formation rate of methyl radical and ethylene as a function of the Mo loading at TOS = 40 min. **c** Normalized concentration of different products *versus* benzene as a function of TOS. Note that the intensity of ethylene signal is divided by 20 times and that of ethane by 10 times. Conditions: 750 °C, photon energy 10.6 eV for detection of methyl radical and ethylene, and 12.0 eV for ethane.

### *Operando* X-ray absorption spectroscopy revealing the feature of Mo single sites

X-ray diffraction (XRD) measurements exhibit typical diffraction patterns of the MWW framework structure[32], and no reflections originating from any Mo species are observed for both fresh and used $x$Mo/MCM catalysts (Supplementary Fig. 7). This hints to a stable MCM-22 zeolite framework and highly dispersed Mo species during the reaction. High-angle annular dark-field scanning transmission electron microscopy (HAADF-STEM) images reveal that the Mo species are atomically dispersed on the MCM-22 zeolite over both the fresh and used $x$Mo/MCM catalysts (Fig. 3a, b and Supplementary Fig. 8). Hence, we further conducted X-ray absorption spectroscopy investigation to unravel the state of the Mo sites under operating conditions. X-ray absorption spectroscopy (XAS) is a powerful technique to elucidate the local environment of atomically dispersed catalyst, however it provides average information[33]. The extended X-ray absorption fine structure (EXAFS) analysis of the fresh $x$Mo/ MCM catalysts show hardly any Mo-Mo backscattering contributions as *e.g.* observed in Mo foil, $MoO_3$ and $Mo_2C$ (Supplementary Fig. 9a), confirming the presence of highly dispersed Mo-species in fresh catalysts. The *operando* X-ray absorption near-edge structure (XANES) spectra in Fig. 3c exhibit an increasing pre-edge intensity upon heating in He, suggesting symmetry transformation of the oxidic Mo species from distorted octahedral to tetrahedral coordination (see Supplementary Fig. 10a for the procedure and Supplementary Fig. 10b for references)[34,35]. *Operando* XANES during reaction at 750 °C showed only slight change of the absorption spectra (Fig. 3d), thereby indicating the highly stable nature of these Mo sites. At the same time, an online mass spectrometer detected formation of $C_2$ hydrocarbons, benzene, and hydrogen (Supplementary Fig. 11). Ex-situ XANES results of used $x$Mo/MCM catalysts are well in line with the *operando* XANES findings (Supplementary Fig. 12). Wavelet transform EXAFS analysis was employed to further understand the structure of Mo species[36,37]. Figure 3e displays only one maximum at $k$ values from 4.0 to 4.5 Å$^{-1}$ for both 0.5Mo/MCM being heated in He and after $CH_4$ conversion. Neither the wavelet

transform feature for the Mo-Mo bond of an Mo foil (8.20 Å$^{-1}$, 2.48 Å not corrected for phase shift), nor that of $Mo_2C$ (8.90 Å$^{-1}$, 2.56 Å not corrected for phase shift) is observed, indicating that Mo species remain Mo single sites, even during MDA reaction. Since the Mo species over both 0.25Mo/MCM and 1Mo/MCM also remain single sites after reaction as shown by HAADF-STEM images (Supplementary Fig. 8) supported by EXAFS (Supplementary Fig. 9b), the linear correlation of the methyl radical concentration with the number of Mo single sites (Fig. 2b) outlines the essential role of Mo single sites in the generation of methyl radicals.

According to the comparison with the Mo-C bond of $Mo_2C$ (3.40 Å$^{-1}$, 1.60 Å not corrected for phase shift) and Mo−O bond of $Na_2MoO_4$ (5.35 Å$^{-1}$, 1.26 Å not corrected for phase shift), the maximum at *ca.* 4–4.5 Å$^{-1}$ observed for 0.5Mo/MCM could be attributed to Mo−O and/or Mo−C[38]. Further quantitative least-squares EXAFS curve-fitting analysis of 0.5Mo/MCM(He) prior to reaction (Supplementary Fig. 13d and Table 2) shows $Mo-O_1$ and $Mo-O_2$ scattering paths in the first coordination sphere[21]. Upon interacting with $CH_4$ (Supplementary Fig. 13f), a new minor satellite peak shows up at 2.2 Å, which may be attributed to Mo−C backscattering of $Mo_2C$ species[22,39] or $Mo-O_4$ backscattering. Coordination numbers of the $Mo-O_2$ and $Mo-C/O_4$ paths were estimated to be 2.1 ± 0.3 and 1.7 ± 0.7, respectively. Supplementary Fig. 14 shows formation of CO during the initial reaction stage, demonstrating reduction of Mo-oxo species, which is quantified to be about 20% according to the CO yield. Further periodic switch pulse reaction using $^{12}CH_4/^{13}CO$ isotope gives benzene with m/z = 79 corresponding to $^{13}C^{12}C_5H_6$ (after subtracting the contribution of the natural abundance), and no hydroformylation products containing oxygen were detected (Supplementary Fig. 15a). It indicates the presence of $^{13}C$ species on the active sites and incorporation into benzene product since $^{13}CO$ can only react with Mo-oxo species. This is consistent with the observation by Gascon et al.[40]. Similarly, benzene isotopologues are also detected during the $^{12}CH_4/^{13}CH_4$ switch pulse reaction (Supplementary Fig. 15b). Therefore, the active site likely contains Mo-CH$_x$ including further coordination to neighboring oxygen in the lattice. We further simulated XANES spectra for all possible

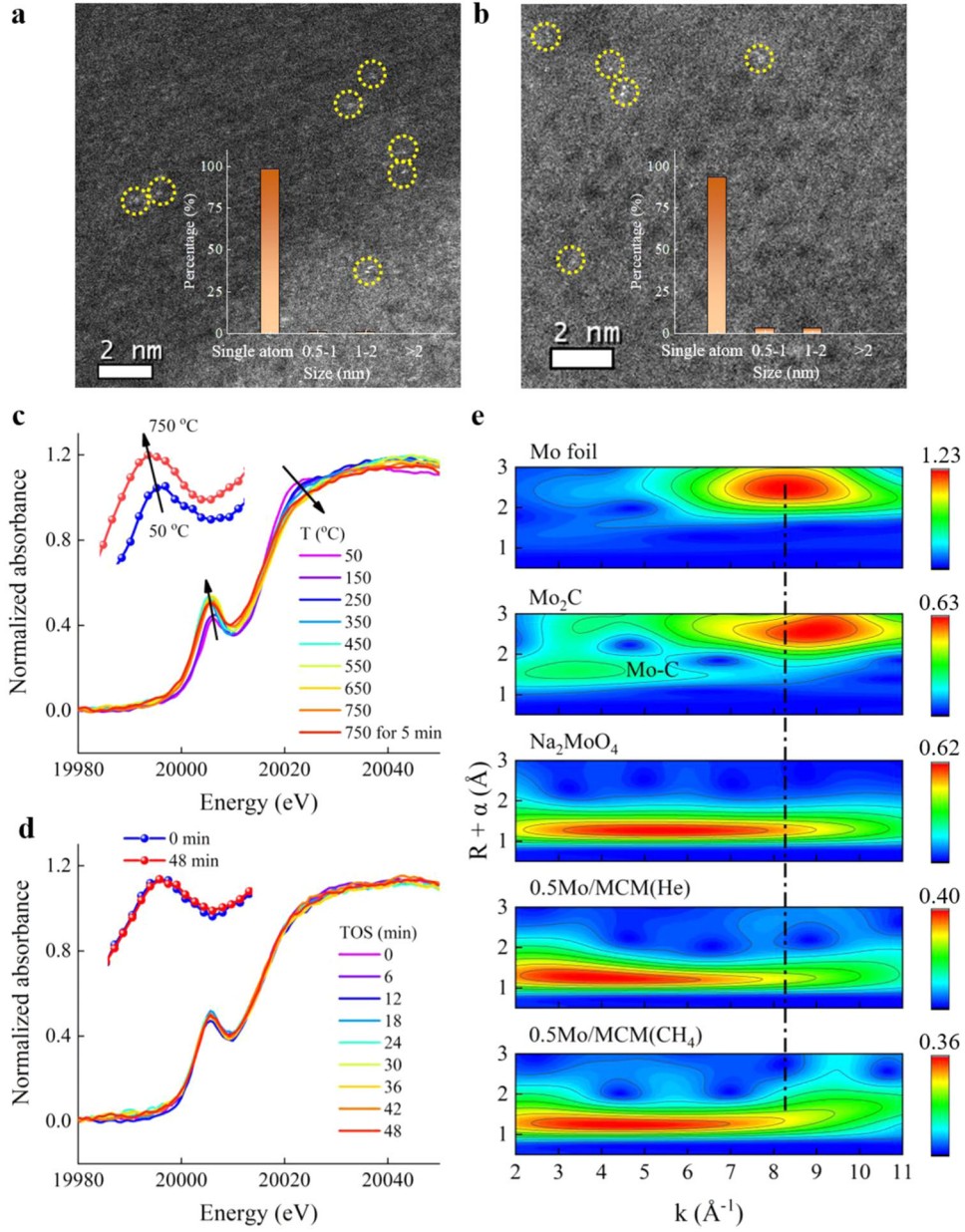

**Fig. 3 | HAADF-STEM and *operando* XAS study of 0.5Mo/MCM. a** STEM image of fresh 0.5Mo/MCM and **b** used 0.5Mo/MCM after reaction at 750 °C for 10 h with the yellow circles highlighting Mo single sites and the inset showing the Mo particle size distribution (statistical number > 100). **c** Mo K-edge XANES spectra during heat-up to 750 °C in He. **d** Mo K-edge XANES spectra during MDA reaction at 750 °C. **e** $k^2$-weighted wavelet transform EXAFS of 0.5Mo/MCM(He) and 0.5Mo/MCM(CH₄) in comparison to Mo foil, bulk Mo₂C and Na₂MoO₄ references.

density functional theory (DFT) optimized structures of Mo-CH$_x$, including O=Mo=O, Mo-CH, Mo-CH₂, Mo-CH₃, O=Mo–C, O=Mo-CH, O=Mo-CH₂, and O=Mo-CH₃ using the FEFF 9.6.4 code for multiple-scattering (MS) calculations of the spectra[41] (Supplementary Fig. 16). The feature of O=Mo-CH₂ (Supplementary Fig. 16h) resembles that of O=Mo=O to some degree (Supplementary Fig. 16a). DFT calculations (Supplementary Fig. 17) confirm that the O=Mo-CH₂ single site is thermodynamically the most stable among all Mo single sites O=Mo-CH$_x$ (x = 0, 1, 2, and 3). The linear combination fitting by taking the 20% reduced species into consideration gives a spectrum (Supplementary Fig. 18) similar to the experimental one in Fig. 3d. Therefore, we propose O=Mo-CH₂ entity acting as the active site (Supplementary Fig. 19) although it is not clear yet the role of the remaining O=Mo=O sites in the reaction, which shall be investigated in the future.

## Reaction pathways over Mo single sites in comparison to Mo₂C nanoparticles

Our previous DFT calculations revealed that methyl radicals readily went through C-C coupling in the gas-phase forming ethane, which further dehydrogenated giving ethylene in the MTOAH reaction[23]. Schwarz et al. also reported that coupling of methyl radicals in the gas phase to C₂ hydrocarbons was indeed possible in their study on methane oxidation[42]. Furthermore, Razdan et al. concluded that ethane was the sole primary product of MDA reaction catalyzed by Mo/H-ZSM-5, and was sequentially dehydrogenated to ethylene and acetylene, which further aromatized to benzene[43]. Figure 2c and Supplementary Fig. 20 show that the ethane concentration ($m/z$ = 30) detected by SVUV-PIMS at a photon energy of 12.0 eV follows a similar profile as that of ethylene, but is overall lower than that of ethylene

(Fig. 2c). Therefore, ethane is most likely the primary product of methyl radicals, as it readily dehydrogenates forming ethylene under reaction conditions (Supplementary Fig. 21), similar to MTOAH reaction[23]. Both the ethane and ethylene concentration decrease while that of benzene increases with TOS during the induction period of 40 min (Fig. 2c), suggesting that benzene is the secondary reaction product. The above results strongly support that MDA and MTOAH processes are closely related.

To study the role of the zeolite, 0.5 wt.% Mo was dispersed on H-ZSM-5 following the same method as 0.5Mo/MCM and denoted as 0.5Mo/ZSM (Supplementary Figs. 22 and 23). It gives a slightly lower $CH_4$ conversion than 0.5Mo/MCM, probably owing to faster mass transfer within the nanosheet-morphology of the MCM-22 zeolite. *Operando* and ex-situ XAS, as well as HAADF-STEM (Supplementary Figs. 9a and 24–26) suggested that the Mo species remain highly dispersed as single sites during the reaction and exhibit a similar coordination configuration as 0.5Mo/MCM (Supplementary Table 3). Furthermore, online SVUV-PIMS detects formation of methyl radicals during the reaction (Supplementary Figs. 27 and 28). Thereby, the results evidence that Mo single sites of 0.5Mo/ZSM also follow methyl radical chemistry.

In order to gain further insights into the role of Mo single sites and the support, we deposited 0.5 wt.% Mo on a $SiO_2$ support by a flame spray pyrolysis method (denoted as $0.5Mo/SiO_2$, Supplementary Fig. 29). HAADF-STEM and EXAFS unraveled that molybdenum is well dispersed in the form of Mo single sites in the fresh $0.5Mo/SiO_2$ catalyst (Supplementary Figs. 30 and 31). The products of non-oxidative methane conversion are dominated by ethylene with a selectivity of about 51% and aromatics selectivity of about 37% at the same reaction conditions as applied for 0.5Mo/MCM (Supplementary Fig. 32). The product distribution for the flame-made sample $0.5Mo/SiO_2$ is similar to MTOAH catalyzed by the single site Fe©$SiO_2$ catalyst[23]. Since methyl radicals were also detected in the gas phase during the reaction by the online SVUV-PIMS (Supplementary Fig. 28), the methyl radical chemistry is very likely a general feature of metal single sites under non-oxidative reaction conditions. However, the normalized methyl radical concentration detected by SVUV-PIMS for $0.5Mo/SiO_2$ is lower than for 0.5Mo/MCM (Supplementary Fig. 28). The *operando* EXAFS spectrum exhibits Mo-Mo scattering at about 2.8 Å over $0.5Mo/SiO_2(CH_4)$ after reaction, indicating aggregation of the Mo sites (Supplementary Fig. 31f, not corrected for the phase shift). HAADF-STEM confirmed the presence of several *ca.* 2 nm sized nanoparticles with β-$Mo_2C$ crystal fringe[44], whilst most Mo species remain single sites (Supplementary Fig. 33). No reflections of Mo species can be detected with XRD, most likely due to the low metal loading (Supplementary Fig. 34). The experiments on $0.5Mo/SiO_2$ revealed a remarkably lower catalytic activity of $Mo_2C$ particles in the generation of methyl radicals, since the Mo loading for $0.5Mo/SiO_2$ and 0.5Mo/MCM is very similar (Supplementary Table 1).

This hypothesis is further validated by increasing the Mo loading of Mo/MCM above 2.0 wt.% (corresponding to 231.4 Mo μmol/$g_{cat}$). For instance, the *operando* XAS analysis of 4Mo/MCM clearly showed the presence of aggregated Mo species (Supplementary Fig. 35) and HAADF-STEM demonstrates the co-existence of Mo single sites and $Mo_2C$ nanoparticles after reaction (Supplementary Figs. 36 and 37). Likewise, the concentration of methyl radicals decreases drastically and no longer linearly increases with the Mo loading up to 4.0 wt.% (Supplementary Fig. 28). These results further support the essential role of Mo single sites in methyl radical chemistry. Agote-Arán et al. also reported that agglomerated Mo species led to a lower catalytic activity over Mo/Silicalite-1 in comparison to Mo/ZSM-5[35]. Ma et al. previously concluded that aggregated $Mo_2C$ nanoparticles could be active in methane activation but mainly led to carbon deposition[16]. This is consistent with the observation of significantly increased carbon deposition over 4Mo/MCM compared to 1Mo/MCM[45]. Therefore,

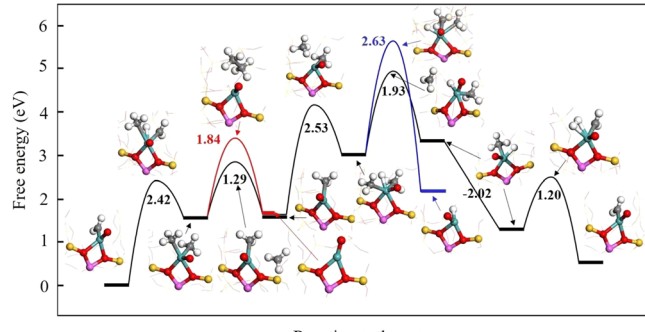

**Fig. 4 | The free energy diagram of $CH_4$ activation and conversion to $CH_3CH_3$ via methyl radical in comparison to surface reaction pathway over the O=Mo–$CH_2$ single site.** The inserts are the transition state structures with the white, gray, red, pink, yellow and green spheres representing H, C, O, Al, Si and Mo atoms, respectively.

the reaction over $Mo_2C$ nanoparticles most likely takes place through a surface reaction pathway and the activated methane participates in C-C coupling on the surface, similar to that over iron nanoparticles in the MTOAH reaction, which also gave mainly carbon deposition[23].

This is further validated by DFT calculations. As depicted in Fig. 4, the first $CH_4$ molecule dissociates over the O=Mo-$CH_2$ single site with a barrier of 2.42 eV. In this process, $CH_2^*$, rather than the Mo atom, plays the role of hydrogen receptor although Mo-H forms in the transition state. Then $CH_3^*$ desorbs as $CH_3$• by overcoming a barrier of 1.29 eV. This is more kinetically favored than the surface C-C coupling of two $CH_3^*$ as ethane (1.84 eV). The dissociation of a second $CH_4$ molecule on the site forming a six-coordinated Mo needs to overcome a barrier of 2.53 eV. The energy barrier for the desorption of one $CH_3^*$ as a $CH_3$• is 1.93 eV in comparison to 2.63 eV for the desorption of $CH_3CH_3^*$ (2.63 eV). Furthermore, the coupling of two $CH_3$• in the gas phase is a strongly exothermic process ($\Delta G = 2.02$ eV) with no kinetic barrier, which leads to formation of ethane. Finally, the Mo-bonded H atom couples with the other H atom from dehydrogenation of $CH_3^*$, leading to formation of a $H_2$ molecule with a barrier of 1.20 eV. Thus, the catalytic cycle is completed and the active site is liberated returning to the initial configuration. The above results demonstrate that formation of methyl radicals is kinetically more favored than surface C−C coupling that forms ethane. For comparison, we also studied methane activation over $Mo_2C$ nanoparticles by taking a $Mo_2C$ slab as a model. One sees from Supplementary Fig. 38 that the surface coupling of two $CH_3^*$ species forming ethane needs to overcome a barrier of 2.66 eV, whereas the barrier to form a gas-phase $CH_3$• radical is much higher (3.74 eV). These results validate that methyl radical chemistry is more favored than the surface reaction pathway over Mo single sites whereas the surface C-C coupling is energetically favored over aggregated Mo species during non-oxidative activation of methane.

To further understand the relationship between MDA and MTOAH, $0.5Mo/SiO_2$ was mixed with H-MCM-22 zeolite at a mass ratio of 1:1 (denoted as 0.5Mo-PM). For comparison, $0.5Mo/SiO_2$ was mixed with an equivalent amount of quartz to keep the reaction under the same space velocity and the resulting catalyst was denoted as 0.5Mo/$SiO_2$-Q. Figure 5 shows that the $CH_4$ conversion rate for 0.5Mo-PM is enhanced and the product distribution is shifted from $C_2$ hydrocarbons toward aromatics, compared to 0.5Mo/$SiO_2$-Q. The shifted product distribution becomes more significant when Mo species sit directly on MCM-22, i.e. 0.5Mo/MCM, as the aromatics selectivity increases to up to 81%, whilst the $C_2$ selectivity decreases to 19%. These results further validate that MDA and MTOAH essentially follow a similar methane activation pathway of methyl radical chemistry, which could be a general feature for metal single sites. The reaction is steered towards aromatization by MCM-22, confirming the bifunctional nature

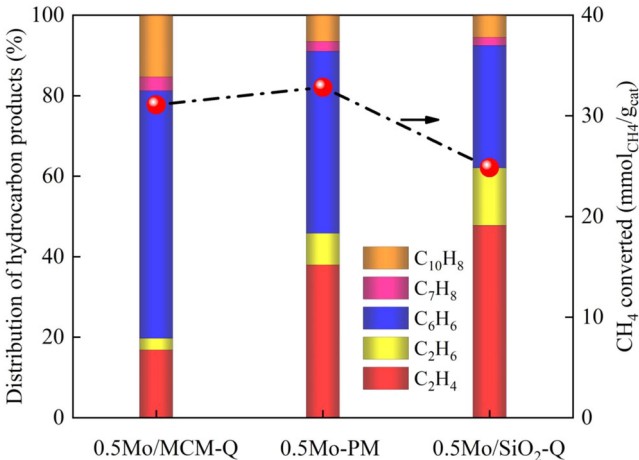

**Fig. 5 | The role of H-MCM-22 in methane conversion, demonstrated by different packing modes of Mo/SiO$_2$ and H-MCM-22.** 0.5Mo/MCM-Q represents the pellet-mixed 0.5Mo/MCM and quartz; 0.5Mo-PM stands for the ball mill-mixed 0.5Mo/SiO$_2$ and H-MCM-22; 0.5Mo/SiO$_2$-Q for the pellet-mixed 0.5Mo/SiO$_2$ and quartz. Note that the mass ratio of the two components in the physical mixture is 1:1 and the left Y-axis represents the distribution of cumulative hydrocarbon products. Reaction conditions: 750 °C, CH$_4$:N$_2$ = 9:1, 0.1 MPa, 0.75 L/(g$_{cat}$·h) and data obtained after 10 h on stream.

of Mo/MCM for MDA. The catalytic activity of the parent H-MCM-22 in aromatization of ethylene/ethane is further verified by catalytic activity studies in Supplementary Fig. 39, which agree with previous studies[46,47].

## Discussion

The mechanism for direct non-oxidative conversion of methane has been investigated for Mo species dispersed over H-MCM-22, H-ZSM-5, and SiO$_2$ with varying Mo loading. *Operando* XAS and SVUV-PIMS combined with HAADF-STEM and DFT calculations reveal the characteristic methyl radical chemistry over Mo single sites, which is followed by the gas-phase C-C coupling forming ethane/ethylene. The products formed through this catalytic methane conversion are dominated by ethylene. This is well in line with the direct methane conversion over the SiO$_2$ lattice-confined Fe single sites, demonstrating that methyl radical chemistry is likely a general feature of metal single sites, regardless of the support. In comparison, the presence of zeolites, either in a physical mixture or as a support, steers the subsequent reaction pathway of ethane/ethylene towards aromatization, giving mainly aromatics as products. However, the aggregated Mo species, i.e. Mo$_2$C nanoparticles, are inactive in methyl radical generation and the reaction most likely is facilitated through surface C−C coupling resulting in carbon deposition. These fundamental findings provide new insights into the mechanism of the direct non-oxidative methane conversion and can lead to further catalyst development.

## Methods
### Catalyst preparation
A series of Mo/H-MCM-22 catalysts with a Mo loading varying between 0.25 to 4.0 wt.%, and 0.5 wt.%Mo/H-ZSM-5 were prepared using an incipient wetness impregnation method. H-MCM-22 and H-ZSM-5 zeolites were purchased from Nankai University. Ammonium heptamolybdate tetrahydrate ((NH$_4$)$_6$Mo$_7$O$_{24}$·4H$_2$O, >99.0%) was used as a precursor. The impregnated zeolite was dried at 105 °C for 12 h and then calcined in air at 500 °C for 5 h. As a reference, a single site 0.5 wt.% Mo/SiO$_2$ catalyst was prepared using a flame spray pyrolysis method. 0.255 g Bis(acetylacetonato)dioxomolybdenum(VI) (MoO$_2$(acac)$_2$, ~99%, Alfa Aesar) and 52.056 g tetraethyl orthosilicate (TEOS, 99.999%, VWR chemicals) were dissolved and diluted in methanol under

ultrasonic dispersion. The precursor solution was fed through a capillary into a CH$_4$/O$_2$ flame (750 mL/min CH$_4$ and 1600 mL/min O$_2$) using a syringe pump (World Precision Instruments) and dispersed by oxygen (5 L/min). A cylindrical steel vessel with a glass fiber filter (24 cm diameter, Whatman GF6) was placed above the flame. The produced particles were collected on this filter with the aid of a vacuum pump[48].

### Catalyst characterization
X-ray diffraction (XRD) was measured by a PANalytic X'Pert Pro-1 instrument using Cu Kα radiation (0.1541 nm). High-angle annual dark-field scanning transmission electron microscopy (HAADF-STEM) was performed on a JEM ARM300F microscope operated at 300 kV. For NH$_3$ temperature-programmed desorption (NH$_3$-TPD) characterization, 50 mg sample was pretreated in pure Ar (30 mL/min) at 500 °C for 1 h, followed by NH$_3$ adsorption at 50 °C for 1 h. After being purged with Ar to remove the physically adsorbed NH$_3$, the sample was heated from 50 to 750 °C at a ramp rate of 10 °C/min in a flowing Ar (30 mL/min) while the NH$_3$ signal was recorded using a Pfeiffer Omnistar GSD 320 mass spectrometer.

### Non-oxidative methane conversion
The reaction was carried out at 750 °C, 0.1 MPa, CH$_4$:N$_2$ = 9:1, and a gas hourly space velocity (GHSV) of 1.5 L/(g$_{cat}$·h) in a quartz tube fixed-bed flow reactor unless otherwise stated. The catalyst was heated from room temperature to 750 °C at a ramp rate 10 °C/min in Ar (30 mL/min). A pre-mixed 90 vol.%CH$_4$/N$_2$ was used as the feed with N$_2$ as an internal standard for the online GC analysis (7890B Agilent GC), which was equipped with TCD (Hayesep Q column) and FID (HP-1 capillary column) detectors. CH$_4$ conversion, product selectivities, and yields were calculated following the previously reported methods[8,9].

### *Operando* X-ray absorption spectroscopy
XAS (including X-ray absorption near-edge structure, XANES, and extended X-ray absorption fine structure, EXAFS) at the Mo K-edge (20,000 eV) were performed at the P65 beamline of the PETRA III synchrotron radiation source (DESY, Hamburg) in transmission mode using ionization chambers. The source of X-rays is an 11 period undulator providing a photon flux of about 10$^{11}$ photons per second. The energy of X-ray photons were further selected by a Si(311) double-crystal monochromator and the beam size was set with slits to 1.0 × 0.2 mm$^2$. XANES spectra were recorded in rapid continuously scanning mode and EXAFS spectra were acquired in step scanning mode. For *operando* XAS measurement, *ca.* 2 mg of 100–200 µm sieved sample was loaded in a quartz capillary microreactor with an outer diameter of 1.5 mm and a wall thickness of 0.02 mm, was fixed between two quartz wool plugs and heated by a self-built high-temperature *operando* XAS cell (Supplementary Fig. 40)[49]. The feed gas was introduced with mass flow controllers (Bronkhorst) at a space velocity of 30,000 h$^{-1}$ and products were analyzed using a mass spectrometer (Pfeiffer Omnistar GSD 320 T). The spectra were normalized and the background subtracted using the ATHENA program from the Demeter software package[50]. The k$^2$-weighted EXAFS functions were Fourier transformed in the k range of 2–11 Å$^{-1}$ and multiplied by a Hanning window with sill size of 1 Å$^{-1}$. The structure refinement was performed in R-space using the ARTEMIS software (Demeter)[50]. For this purpose, the corresponding theoretical backscattering amplitudes and phases were calculated by FEFF 6. The theoretical data were then adjusted to the experimental spectra by a least square method[41] in R-space between 1 and 3 Å. First, the amplitude reduction factor (S$_0^2$ = 0.94) was calculated using the Mo foil reference spectrum, then relative mean-square displacement of the atoms included in the path (Debye-Waller factor, σ$^2$) were fixed to values derived from fitting corresponding references (Supplementary Fig. 41 and Table 4). The number of identical paths (CN), change in the half path

length ($\Delta R$), energy shift of the path ($\Delta E_0$) were then refined with $Na_2MoO_4$ and $Mo_2C$ crystal structures. Morlet wavelets of constant shape were used to generate continuous wavelet transformation plots using HAMA software[51]. The function f(a,b) is produced by the continuous wavelet transform of an EXAFS signal, and plotted on ($k$, R+$\alpha$) axes[52].

$$f(a,b) = \int \chi(k)\psi(ak+b)dk \tag{1}$$

Morlet wavelet parameters $\kappa$ and $\sigma$ define the shape of the mother wavelet $\psi$ ($\kappa$ represents the number of oscillations, and $\sigma$ is the half-width of the wavelet envelope)[53]. In this work, wavelet transforms were performed over the regions 0.5–3.0 Å using the wavelet parameters $\kappa = 5$ and $\sigma = 1$[54].

### Synchrotron vacuum ultraviolet photoionization mass spectroscopy

SVUV-PIMS was performed at the National Synchrotron Radiation Laboratory (NSRL, Hefei, China). Synchrotron radiation from an undulator beamline (BL03U) was monochromatized with a 200 lines/mm laminar grating, and this grating covers the photon energy from 7.5 to 22 eV with an energy resolving power of 3000 (E/$\Delta$E @ 10 eV)[55]. 200 mg sample was loaded in a horizontal quartz reactor (o.d. 10 mm, i.d. 7 mm, L. 480 mm, Supplementary Fig. 42). The pressure of the catalytic reactor was maintained at 0.47 kPa, similar to the reported studies on radicals in oxidative coupling of methane, propane oxidative dehydrogenation, and methane oxybromination reactions, and non-oxidative conversion of methane[27,29,30,56]. Prior to reaction, the catalyst was heated from room temperature to 750 °C in Ar (40 mL/min). A small fraction of the products from the catalyst bed was sampled through a quartz nozzle (i.d. 450 μm) into the ionization chamber[57]. The molecular beam was crossed and ionized by the synchrotron VUV light and mass-analyzed by a homemade orthogonal time-of-flight mass spectrometer (TOF-MS)[58]. The ratio of mole fraction between $CH_3\bullet$ and $C_2H_4$ with PIMS was calculated as follows[59],

$$\frac{x(CH_3\cdot)}{x(C_2H_4)} = \frac{A(CH_3\cdot)}{A(C_2H_4)} \div \frac{\sigma(CH_3\cdot)}{\sigma(C_2H_4)} \div \frac{D(CH_3\cdot)}{D(C_2H_4)} \tag{2}$$

where A($CH_3\bullet$) and A($C_2H_4$) were the respective integrated peak areas of $CH_3\bullet$ and $C_2H_4$ at the photon energy of 10.6 eV; $\sigma(CH_3\bullet)$ and $\sigma(C_2H_4)$ were the respective photoionization cross-sections of $CH_3\bullet$ and $C_2H_4$ at the photon energy of 10.6 eV; D($CH_3\bullet$) and D($C_2H_4$) were the respective mass discrimination factors of $CH_3\bullet$ and $C_2H_4$. At the combustion station of NSRL, the mass discrimination factor was determined as follows[60],

$$D = \left(\frac{x}{30}\right)^{0.76155} \tag{3}$$

where $x$ denotes m/z (15 for methyl radical, 28 for ethylene, and 78 for benzene).

The photoionization cross-sections at the photon energy of 10.6 eV of methyl radical, ethylene, and benzene were 6.90, 3.50, and 29.36, respectively[61,62]. The mass discrimination factors were 0.59, 0.95, and 2.07 for methyl radical, ethylene and benzene, respectively.

### Density functional theory calculations

All the calculations based on density functional theory (DFT) were performed using Vienna ab initio simulation package (VASP) in which the projector augmented wave method was adopted[63–65]. The revised-PBE under generalized gradient approximation (GGA) was chosen as the exchange-correlation functional[66–68]. The DFT-D3 method was used in dispersion corrections[69]. A plane-wave energy cutoff of 400 eV was used in all the energy calculations. The convergence thresholds of electronic energies and maximum atomic forces are set as $10^{-5}$ eV and 0.05 eV/Å, respectively. A complete periodic cell of MCM-22 structure was used as the theoretical model to investigate the MDA mechanism. The Γ-point was adopted for Brillouin zone sampling. The zero-point energy and entropic corrections were calculated by finite displacement method. The climbing-image nudge elastic band (CI-NEB) method included in VTST code was used to locate the transition states (TSs)[70].

## Data availability
All data supporting the findings of this study are available within the paper, its Supplementary Information files, and Supplementary Data 1.

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

## Acknowledgements

This work was supported by the National Natural Science Foundation of China (Grant nos. 21761132035, X.P., 21805300, X.H.), DFG-grant (GR3987/9-1, J.G.), State Key Laboratory of Catalysis in DICP (Grant no. N-21-07, X.P.), Fundamental Research Program of Shanxi Province (Grant nos. 202203021221042, X.H. and 202204021301010, X.H.), Shanxi Province Science and Technology Innovation Talent Team (Grant no. 202204051002025, X.H.), and State Key Laboratory of Coal and CBM Co-mining (Grant no. 2022KF12, X.H.). We acknowledge the light source PETRA III at DESY where *operando* XAS was carried out, and the beamline BL14W1 and BL13SSW at Shanghai Synchrotron Radiation Facility, and the beamline of BL03U at National Synchrotron Radiation Laboratory. We thank Edmund Welter and Ruidy Nemausat for their kind assistance in using beamline P65 and Dr. Ruifang Liu for her assistance with the TOC artwork.

## Author contributions

X.H. contributed to sample preparation, characterization, reaction test and writing. D.E. conducted Mo/SiO$_2$ preparation, *operando* XAS measurement, and data analysis. B.S., A.Z., A.G., and J.-D.G. contributed to the development of Mo/SiO$_2$ FSP preparation, to data analysis, and writing up of the corresponding results. G.Q. and J.X. carried out DFT calculations. J.Y. and Y.P. assisted the SVUV-PIMS experiment and data analysis. M.L. did HAADF-STEM measurement. J.H. and X.G. participated in discussion of the reaction results. H.Y. assisted a part of the reaction tests. X.P. and X.B. guided the work and wrote the manuscript. All authors commented on the final manuscript.

## Competing interests

The authors declare no competing interests.
