## [Peer Review File · Nature Communications]

REVIEWER COMMENTS

Reviewer #1 (Remarks to the Author):

The authors performed a great revision of the paper once again, with some more experimental data.

I can better understand their explanation for virtually identical XANES spectra of Mo recorded during the reduction of 0.5Mo/MCM-22. Am I correct in understanding that this phrase - “The dramatically decreasing CO yield implies that the rest O=Mo=O sites could have been deactivated by carbonaceous species before reduction” - basically means that as much as 60% of Mo-centers are not active during the reaction. If so, the authors should make it very clear in the text, because if not all Mo-centers are active it would somewhat contradict the linear Mo-loading dependence of benzene formation rate on Fig. 1b and formation of CH₃ radicals on Fig. 2. Also, if majority of Mo-sites are deactivated even before reduction, than operando XAS data on Fig .3 is not very representative of the active phase.

Furthermore, while isotope labelling experiments are very nice, the most likely reason for the isotope exchange are not the carbon atoms on Mo-centers, but exchange with the hydrocarbons confined inside the zeolite pores as shown previously (doi: 10.1021/acscatal.8b02491). I note here that under applied conditions CO can get involved into the formation of aromatic molecules through, for example, hydroformylation reactions.

Reviewer #2 (Remarks to the Author):

Most of my concerns were addressed properly and as a result the manuscript has improved well. However it seems like Figure S5 is still in the SI, but not discussed in the manuscript. The authors should remove not discussed figures.

Some minor comments:

1) The switch pulse experiments are only analysed for C6, why not for C2?

2) Concerning the fitting procedure of the XANES data with DFT calculations. These do not agree so well with the experimental data. The authors should comment on this and maybe relate to papers, where similar fittings were done.

Well written manuscript, which deserves publications in Nat. Commun.

Point-by-point response to the reviewers' comments

Reviewer #1

Comment 1:

The authors performed a great revision of the paper once again, with some more experimental data.

(a) I can better understand their explanation for virtually identical XANES spectra of Mo recorded during the reduction of 0.5Mo/MCM-22. Am I correct in understanding that this phrase - “The dramatically decreasing CO yield implies that the rest O=Mo=O sites could have been deactivated by carbonaceous species before reduction” - basically means that as much as 60% of Mo-centers are not active during the reaction. If so, the authors should make it very clear in the text, because if not all Mo-centers are active it would somewhat contradict the linear Mo-loading dependence of benzene formation rate on Fig. 1b and formation of CH₃ radicals on Fig. 2. Also, if majority of Mo-sites are deactivated even before reduction, than operando XAS data on Fig. 3 is not very representative of the active phase.

(b) Furthermore, while isotope labelling experiments are very nice, the most likely reason for the isotope exchange are not the carbon atoms on Mo-centers, but exchange with the hydrocarbons confined inside the zeolite pores as shown previously (doi: 10.1021/acscatal.8b02491). I note here that under applied conditions CO can get involved into the formation of aromatic molecules through, for example, hydroformylation reactions.

Response: We appreciate the reviewer's comments about our work. We divide the question into two points:

(a) “The dramatically decreasing CO yield implies that the rest O=Mo=O sites could have been deactivated by carbonaceous species before reduction” in ESI is changed to “It was estimated that ~20% Mo species have been reduced according to the CO yield determined by GC. However, it is not clear yet the role of the remaining O=Mo=O sites in the reaction, which shall be investigated in the future”. The periodic isotope switch pulse experiments (Supplementary Fig. 15) confirmed the

transformation of O=Mo=O to O=Mo-CH₂, which was consistent with a previous study [Chem. Sci., 2018, **9**, 4801-4807]. Lezcano-González *et al.* suggested that isolated Mo-oxo species after calcination were converted into metastable MoC_xO_y species by CH₄, which were primarily responsible for C₂H_x/C₃H_x formation [Angew. Chem. Int. Ed., 2016, **55**, 5215-5219]. Our *operando* XANES during reaction at 750 °C showed little change at the Mo K-edge (Fig. 3d), which implied transformation of a small fraction of O=Mo=O species to O=Mo-CH₂. Kosinov *et al.* also reported partially reduced single-atom Mo sites stabilized by the zeolite framework over Mo/ZSM-5 [Angew. Chem. Int. Ed., 2018, **57**, 1016-1020].

The periodic ¹²CH₄/¹³CO (and ¹²CH₄/¹³CH₄) switch pulse experiments over 0.25Mo/MCM and 1Mo/MCM catalysts were also provided, as shown in Figure R1. Figure R1c shows that the ¹³C incorporation degree in benzene increased linearly with the Mo loading. It demonstrates again a linear dependence of the number of active O=Mo-CH₂ sites on the Mo-loading since ¹³CO can only react with the Mo oxides and the resulting ¹³C in benzene products is associated with Mo active sites. Therefore, the linear relationship in Figure 1b and Figure 2b is reasonable.

Figure R1. (a) Periodic $^{12}\text{CH}_4/^{13}\text{CO}$, (b) $^{12}\text{CH}_4/^{13}\text{CH}_4$ switch pulse experiment over the $x\text{Mo/MCM}$ catalysts, and (c) ^{13}C incorporation degree in ^{12}C labelled benzene as a function of Mo loading. Reaction conditions: 200 mg catalyst, 750 °C, 0.1 MPa, carrier gas: 15 mL/min of Ar, pulse gas A: $^{12}\text{CH}_4$, pulse gas B: 2.5 vol.% $^{13}\text{C}^{18}\text{O/He}$, pulse gas C: $^{13}\text{CH}_4$, pulse is allowed 5 mL every 6 min.

(b) We agree with the reviewer that the hydrocarbons confined inside the zeolite pores may take part in benzene formation, as reported by Hensen and coworkers, namely “hydrocarbon pool mechanism” [*Angew. Chem. Int. Ed.*, 2018, **57**, 1016-1020; *ACS Catal.*, 2018, **8**, 8459-8467]. The article is cited as [22].

The periodic $^{12}\text{C}_2\text{H}_6/^{13}\text{CO}$ isotope switch pulse reaction over pure H-MCM-22 zeolite (Figure R2) did not show $^{13}\text{C}^{12}\text{C}_5\text{H}_6$ formation (after subtracting the contribution of the natural abundance) by contrast to formation of $^{13}\text{C}^{12}\text{C}_5\text{H}_6$ over Mo/HMCM-22. It indicates the essential role of Mo species during MDA reaction. Therefore, only the carbon species associated with the Mo-sites, i.e. Mo- ^{13}C , can be incorporated into aromatics products.

No hydroformylation products containing oxygen were detected during the $^{12}\text{CH}_4/^{13}\text{CO}$ periodic switch pulse reaction, suggesting that there may not be hydroformylation reaction.

Figure R2. The periodic $^{12}\text{C}_2\text{H}_6/^{13}\text{CO}$ isotope switch pulse reaction over pure H-MCM-22 zeolite. Reaction conditions: 200 mg of H-MCM-22, 750 °C, 0.1 MPa, carrier gas: 20 mL/min of Ar, pulse gas A: 20 mL/min of 4 vol.% $^{12}\text{C}_2\text{H}_6/\text{He}$, pulse gas B: 20 mL/min of 2.5 vol.% $^{13}\text{CO}/\text{He}$, and pulse volume: 5 mL every 10 min.

Razdan *et al.* concluded that ethane was the sole primary product of MDA reaction catalyzed Mo/H-ZSM-5 [*J. Catal.*, 2020, **381**, 261-270]. Hence, the periodic

$^{13}\text{C}_2\text{H}_6/^{12}\text{C}_2\text{H}_6$ isotope switch pulse reaction over 0.5Mo/MCM and pure H-MCM-22 zeolite were conducted. The results showed that the isotope exchange behavior of ethane dehydroaromatization over 0.5Mo/MCM was very similar with that of MDA, as shown in Figure R3. The ^{13}C incorporation degree in benzene over 0.5Mo/MCM was ca. 5.6 times higher than that over pure H-MCM-22 zeolite, suggesting the essential role of Mo species again.

Figure R3. The periodic $^{13}\text{C}_2\text{H}_6/^{12}\text{C}_2\text{H}_6$ isotope switch pulse reaction over (a) 0.5Mo/MCM, and (b) pure H-MCM-22 zeolite. Reaction conditions: 200 mg of sample, 750 °C, 0.1 MPa, carrier gas: 20 mL/min of Ar, pulse gas A: 20 mL/min of $^{13}\text{C}_2\text{H}_6$, pulse gas B: 20 mL/min of 4 vol.% $^{12}\text{C}_2\text{H}_6/\text{He}$, and pulse volume: 5 mL every 6 min.

Since the mechanism of C_2 to benzene mechanism on single-site Mo/H-MCM-22 is out of the scope of this paper, the data in Figures R1-R3 will be submitted in a separate paper for publication.

Reviewer #2

Most of my concerns were addressed properly and as a result the manuscript has improved well. However it seems like Figure S5 is still in the SI, but not discussed in the manuscript. The authors should remove not discussed figures.

Some minor comments:

1) The switch pulse experiments are only analysed for C₆, why not for C₂?

2) Concerning the fitting procedure of the XANES data with DFT calculations.

These do not agree so well with the experimental data. The authors should comment on this and maybe relate to papers, where similar fittings were done.

Well written manuscript, which deserves publications in Nat. Commun.

Response: We thank the reviewer for the suggestions. Figure S5 was removed.

Comment 1: The switch pulse experiments are only analysed for C₆, why not for C₂?

Response:

Since benzene is the main product of MDA over 0.5Mo/MCM and the selectivity of C₂ is only 4.8%, we mainly analyzed benzene in the switch pulse experiments. Similar experiments were reported previously [*Science*, 2015, **348**, 686-690; *Science*, 2016, **353**, 563-566; *Angew. Chem. Int. Ed.*, 2018, **57**, 1016-1020; *Chem. Sci.*, 2018, **9**, 4801-4807; *ACS Catal.*, 2023, **12**, 1-10].

Comment 2: Concerning the fitting procedure of the XANES data with DFT calculations. These do not agree so well with the experimental data. The authors should comment on this and maybe relate to papers, where similar fittings were done.

Response:

We thank the reviewer for this comment. We showed in Figures S16g, S16h and S16i that the theoretical XANES spectra had all the features that were observed in the experimental spectra. DFT calculations in Figure S17 further validated the most stable structure of O=Mo-CH₂, which is most likely the active site. The actual structure of the Mo single site may contain O=Mo-CH₂ and O=Mo=O, as shown by the linear combination fitting (Figure S18). Similar analysis can be found in previous studies, e.g.

on the Mo K-edge for S-O exchange [*ACS Catal.*, 2019, **9**, 2568-2579].